# Review of Antimicrobial Resistance in Wastewater in Japan: Current Challenges and Future Perspectives

**DOI:** 10.3390/antibiotics11070849

**Published:** 2022-06-24

**Authors:** Hiroaki Baba, Masateru Nishiyama, Toru Watanabe, Hajime Kanamori

**Affiliations:** 1Department of Infectious Diseases, Internal Medicine, Tohoku University Graduate School of Medicine, Sendai 980-8574, Japan; 2Department of Food, Life and Environmental Sciences, Faculty of Agriculture, Yamagata University, Tsuruoka 997-8555, Japan; m-nishiyama@tds1.tr.yamagata-u.ac.jp (M.N.); to-ru@tds1.tr.yamagata-u.ac.jp (T.W.)

**Keywords:** antibiotic-resistant bacteria, antibiotic resistance genes, water environment, wastewater, sewage treatment plant, residual antibiotics, one health approach, Japan

## Abstract

Antimicrobial resistance (AMR) circulates through humans, animals, and the environments, requiring a One Health approach. Recently, urban sewage has increasingly been suggested as a hotspot for AMR even in high-income countries (HICs), where the water sanitation and hygiene infrastructure are well-developed. To understand the current status of AMR in wastewater in a HIC, we reviewed the epidemiological studies on AMR in the sewage environment in Japan from the published literature. Our review showed that a wide variety of clinically important antibiotic-resistant bacteria (ARB), antibiotic resistance genes (ARGs), and antimicrobial residues are present in human wastewater in Japan. Their concentrations are lower than in low- and middle-income countries (LMICs) and are further reduced by sewage treatment plants (STPs) before discharge. Nevertheless, the remaining ARB and ARGs could be an important source of AMR contamination in river water. Furthermore, hospital effluence may be an important reservoir of clinically important ARB. The high concentration of antimicrobial agents commonly prescribed in Japan may contribute to the selection and dissemination of AMR within wastewater. Our review shows the importance of both monitoring for AMR and antimicrobials in human wastewater and efforts to reduce their contamination load in wastewater.

## 1. Introduction

Antimicrobial resistance (AMR), which makes common life-threatening infections difficult or impossible to treat, is continuing to increase globally and now has significant public health and socioeconomic implications. According to recent estimates, if not tackled by 2050, AMR could cause 10 million deaths annually and global economic losses of up 100 trillion USD [1]. In fact, an estimated 4.95 million deaths were already linked to AMR in 2019 [2]. Antibiotic resistance bacteria (ARB) and antibiotic resistance genes (ARGs) with similar biochemical and genetic characteristics to those of humans have also been detected in animals and their environments, suggesting that AMR is circulating in the ecosystem [3]. Therefore, it is necessary to tackle AMR beyond the framework of each relevant field by applying the principle of One Health, in which the health of various fields, such as humans, animals, and environments, interact with each other [4].

In terms of One Health, wastewater is an important reservoir of AMR because it contains various AMR from humans, animals, and environments [5]. The residual antimicrobials in wastewater that are used in humans, animals, and crop plants also are of great concern because they can drive the development and spread of AMR as a result of the selective pressure on bacteria, although the dynamics remain poorly understood [6,7,8,9,10]. A recent metagenomic analysis of ARGs in untreated sewage around the world revealed a higher abundance of ARGs in wastewater in low- and middle-income countries (LMICs) than in high-income countries (HICs), reflecting the state of sanitation and general health [11].

Although the direct discharge of untreated human waste containing many ARB and ARGs into the soil and water environment is one of the major contributors to environmental AMR contamination in LMICs [9,12], in HICs, where the sanitation infrastructure is well-developed, most human waste is released into the water environment (rivers or oceans) after being treated at sewage treatment plants (STPs) [13,14,15,16,17]. STPs were developed to ensure the removal of harmful microorganisms, so they play a pivotal role in reducing AMR [5]. However, because they were not intentionally designed to manage AMR [18], ARB and ARGs can persist even after STP treatment and remain at detectable levels in water environments receiving the discharge [5,9]. To date, the levels of ARB and ARGs that pose a risk to human health are unknown [19], and there is a concern that they may become a source of AMR in water and soil environments through inadequate sewage treatment and sewer overflows [11,16,20,21,22].

In Japan, the population coverage of wastewater treatment systems is 91.4%, which is one of the highest rates in the world [23]; however, wastewater from the remaining approximately 14 million people is still discharged directly into the environment without treatment [23]. Furthermore, 25% of wastewater treatment systems are combined sewer systems that carry both untreated wastewater and rainwater simultaneously, so they have a high risk of causing environmental contamination after heavy rains and flooding, events that have become exacerbated in recent years because of climate change [23].

As in other HICs, in Japan, most of the large centralized STPs use a conventional activated sludge (CAS) process, and most of the small to medium-sized STPs use an oxidation ditches (OD) process, usually combined with chlorination [16,23,24,25]. In STPs, wastewater is treated until it meets National Effluent Standards, which define the permissible limit of coliform bacteria in STP effluent as a daily average of 3000 colonies/cm^3^ [26]; however, no regulations limit the discharge of AMR or antimicrobial residues, and the actual amount of them in STP effluent is still unknown.

To understand the current knowledge about and challenges related to AMR in wastewater in a HIC, we performed a literature review to evaluate the current status of research on AMR in wastewater in Japan. On the basis of the review, we further assessed the risks to humans of AMR and antimicrobials in sewage on humans and the effectiveness of risk mitigation interventions in HICs.

## 2. Materials and Methods

For this review, we focused on AMR in bacteria other than acid-fast bacteria. We excluded articles on non-bacterial microorganisms, including fungi and viruses, and did not consider disinfectant resistance. Although agricultural wastewater and industrial discharge are also considered to be important reservoirs of AMR in other countries [9], our review focused only on sewage because research has shown that AMR in river water in Japan originates mainly from human waste [8].

We conducted a literature search to identify articles written in English or Japanese and published between January 1991 and September 2021. We searched for articles in PubMed (https://pubmed.ncbi.nlm.nih.gov/, accessed on 19 May 2022) and the Igaku Chuo Zashi (ICHUSHI) database, a database of articles written in Japanese (https://search.jamas.or.jp/, accessed on 19 May 2022), with the following keywords: Japan, antimicrobial resistance, wastewater, and sewage (see Appendix A).

The literature search identified 125 articles. We excluded 1 article because the full-text publication was unavailable (Figure 1), and after reviewing the full text of the remaining 124 articles, we excluded another 43 articles that were not related to AMR in sewage in Japan because they described review articles (n = 24) or focused on non-bacterial microorganisms (n = 9), acid-fast bacteria (n = 2), disinfectant resistance (n = 1), livestock wastewater (n = 3), or environmental AMR that are not related to sewage (n = 4). We further excluded review articles (n = 5), clinical case reports (n = 18), and experimental studies (n = 21). The remaining 37 articles were included in the review and categorized into articles on AMR or antimicrobials; in addition, they were further categorized into articles on influent, effluent, and sludge in STPs, hospital effluent, or river water (Figure 1).

## 3. Results and Discussion

### 3.1. Overview of Articles in AMR and Antimicrobials in Wastewater in Japan

The characteristics of the included articles are summarized in Table 1. Of the 37 articles, 26 (70%) were on AMR, 10 (27%) were on antimicrobials, and 1 (3%) was on both. Almost all the articles on AMR in sewage (25 articles, 93%) used culture-based methods, and nearly half (12 articles, 44%) of the articles were focused on *Escherichia coli* as the target bacterial species [27,28,29,30,31,32,33,34,35,36,37,38]. The remaining two articles (7%) reported on the use of consecutive ultrafiltration [39] and metagenomic analysis [40] to directly detect ARGs in water. In total, investigations of ARGs were reported in 18 articles (67%); most (10 articles, 56%) reported on studies that used whole-genome sequencing (WGS) for the comprehensive detection of ARGs [28,32,36,41,42,43,44,45]; 4 (22%), on studies that detected extended-spectrum β-lactamase (ESBL) genes (e.g., *bla*_TEM_, *bla*_SHV_, *bla*_OXA_, *bla*_CTX-M_) by polymerase chain reaction (PCR) [33,34,37,39]; and 3 (17%), on studies that detected tetracycline resistance genes (e.g., *tetA*, *tetB*) [33,34,39]. In Japan, most of the studies on AMR in sewage have been conducted with culture-based methods for detecting ARB, and most have used WGS of detected ARB for detecting ARGs.

Of the 11 articles on antimicrobials, 10 (90%) described studies that used liquid chromatography-mass spectrometry (LC-MS) to determine the concentration of antimicrobial residues in water samples [8,46,47,48,49,50,51,52,53,54], and only 1 article (10%) presented a study that used gas chromatography-mass spectrometry [55]. Macrolides were included in 8 articles [8,46,47,49,50,51,53,54]; sulfonamides/trimethoprim, in 7 [8,47,48,49,50,51,53]; quinolones, in 5 [46,47,49,50,51], and tetracyclines, in 2 [50,51]. Only one article included beta-lactams (ampicillin and amoxicillin only) [48], and none included aminoglycosides. One article evaluated both AMR and antimicrobials in STP influent/effluent, hospital effluent, and river water [46]. The effect of STP wastewater treatment systems on AMR and antimicrobial residues was reported in 14 articles, of which 12 examined only the activated sludge (AS) process (4 examined CAS [27,46,50,52], and 8 examined AS [29,30,37,48,53,54,55,56]); 6, the AS process and chlorination [27,30,37,46,52,56]; and 4, the AS process and ozonation [27,34,46,53]. One article evaluated ultraviolet (UV)-based processes [51]. These results suggest that most studies on antimicrobials in sewage in Japan have evaluated macrolides and sulfonamides/trimethoprim with LC-MS and that the effect of the AS process on residual antimicrobials has been particularly well studied.

**Table 1 antibiotics-11-00849-t001:** Overview of antimicrobial resistance and antimicrobials in wastewater in Japan.

Subject	Source	Author Year	Detection Method	Studied Bacteria	Resistance Genes	Studied Antimicrobials	Wastewater Treatment Systems
AMR	STP influent	Yanagimoto et al. 2020 [57]	Culture-based	*Salmonella* spp.	NA	NA	NA
		Hayashi et al. 2019 [31]	Culture-based WGS*	*Escherichia* *coli*	*aadA1* *aadA2* *aph(3′)-Ia* *bla* _TEM-1_ *catA1* *cmlA1* *dfrA5* *dfrA12* *dfrA17* *mcr-1* *mdf(A)* *mdf(B)* *tet(A)* *tet(B)* *tet(M)*	NA	NA
		Suzuki et al. 2019 [43]	Culture-based WGS	*Klebsiella* *quasipneumoniae*	*aadA1* *arr2* *bla* _CTX-M-2_ *bla* _DHA-1_ *bla* _KHM-1_ *bla* _OXA-10_ *cmlA5* *dfrA14* *qnrB4* *sul1*	NA	NA
		Tanaka et al. 2019 [33]	Culture-based PCR	*Escherichia* *coli*	*armA* *bla* _CTX-M_ *fosA3* *mcr-1* *mcr-2* *mcr-3* *rmtB* *rmtC* *tetA* *tetB* *tetM*	NA	NA
		Ishiguro et al. 2005 [58]	Culture-based PCR	*Salmonella enterica* serovar newport	*bla* _CMY_	NA	NA
	STP effluent	Urase et al. 2020 [30]	Culture-based	*Escherichia coli* CRE	NA	NA	AS+Chrolination Oxidation ditch process +Chrolination
		Sekizuka et al. 2019 [41]	Culture-based WGS	*Aeromonas hydrophila* *Aeromonas caviae*	*aac(6′)-Ia* *aadA2* *bla* _KPC-2_ *bla* _OXA-669_ *bla* _OXA-726_ *bla* _OXA-780_ *bla* _MOX-12_ *cepS* *cphA2* *mcr-3* *mph(A)* *sul1* *tet(E)*	NA	NA
		Sekizuka et al. 2019 [32]	Culture-based WGS	*Escherichia* *coli*	*aadA5* *aph(3′)-Ia* *bla* _CTX-M-55_ *bla* _EC_ *bla* _NDM-5_ *bla* _TEM-135_ *dfrA14* *dfrA17* *floR* *fosA3* *qnrS1* *qnrS2* *tet(A)*	NA	NA
		Sekizuka et al. 2018 [42]	Culture-based WGS	*Klebsiella* *pneumoniae*	*aac(3)-IIa* *aac(3)-IId* *aac(6′)Ib-cr* *aadA2* *aph(3′’)-Ib* *aph(6)-Id* *bla* _KPC-2_ *bla* _OXA-1_ *bla* _SHV-1_ *bla* _TEM-1B_ *catB4* *dfrA12* *fosA3* *fosA6* *qacEdelta1*	NA	NA
	STP sludge	Miura et al. 2013 [45]	Culture-based WGS	*Acidovax* sp.	NA	NA	NA
		Mori et al. 2008 [40]	Metagenomic analysis	NA	*ble*	NA	NA
	Hospital effluent	Eda et al. 2021 [59]	Culture-based WGS	*Klebsiella* *pneumoniae*	*aac(6′)-Ib* *bla* _KPC_ *bla* _OXA-9-like_ *bla* _TEM-1A-like_	NA	NA
		Okubo et al. 2019 [34]	Culture-based PCR	*Escherichia* *coli*	*bla* _TEM_ *bla* _SHV_ *bla* _OXA_ *bla* _CTX-M_ *tetA* *tetB* *tetC* *tetD* *tetE* *tetG*	NA	NA
		Sakagami et al. 1992 [60]	Culture-based	*Staphylococcus* *aureus*	NA	NA	NA
	River water	Liu et al. 2020 [39]	Consecutive ultrafiltration PCR	NA	*bla* _TEM_ *tetA* *tetW* *sul1* *ereA* *qnrD*	NA	NA
		Suzuki et al. 2019 [61]	Culture-based	*Escherichia* *coli*	NA	NA	NA
	STP influent/effluent	Suwa et al. 2015 [37]	Culture-based PCR	*Escherichia* *coli*	*bla* _TEM_ *bla* _SHV_ *bla* _OXA_ *bla* _CTX-M_ *bla* _CMY_	NA	AS Chlorination
	STP influent/effluent/sludge	Honda et al. 2020 [29]	Culture-based	*Escherichia* *coli*	NA	NA	AS
		Furukawa et al. 2015 [56]	Culture-based PCR	*Enterococcus* spp.	*vanA* *vanB*	NA	AS Chlorination
	STP influent River water	Ogura et al. 2020 [28]	Culture-based WGS	*Escherichia* *coli*	*aac3-VIa* *aadA* *bla* _CMY_ *bla* _CTX-M-1_ *bla* _CTX-M-8_ *bla* _CTX-M-9_ *bla* _TEM-1D_ *cmlA* *tetA* *tetB* *tetC*	NA	NA
		Nishiyama et al. 2015 [62]	Culture-based PCR	*Enterococcus* spp.	*vanA* *vanB* *vanC1* *vanC2/C3*	NA	NA
	STP effluent Hospital effluent	Gomi et al. 2018 [44]	Culture-based WGS	CPE	*bla* _GES-5_ *bla* _GES-6_ *bla* _GES-24_ *bla* _IMP-8_ *bla* _IMP-19_ *bla* _KPC-2_ *bla* _NDM-5_ *bla* _VIM-1_	NA	NA
		Gomi et al. 2017 [36]	Culture-based WGS	*Escherichia* *coli*	*bla* _CTX-M_ *bla* _SHV_ *bla* _TEM_	NA	NA
	STP effluent River water	Iwane et al. 2001 [38]	Culture-based	*Escherichia* *coli*	NA	NA	NA
	STP effluent Hospital effluent River water	Yamashita et al. 2017 [35]	Culture-based	*Escherichia* *coli*	NA	NA	NA
	STP influent/effluent Hospital effluent River water	Azuma et al. 2021 [27]	Culture-based	CRE ESBL MDRA MDRP MRSA VRE	NA	NA	CAS CAS+Chlorination CAS+Ozonation Solar irradication
Anti- microbials	STP effluent	Kim et al. 2009 [51]	LC-MS/MS	NA	NA	Azithromycin Clarithromycin Erythrimycin Levofloxacin Nalidixic acid Norfloxacin Sulfadimethoxine Sulfamethoxazole Tetracycline Chlorotetracycline Lincomycin Trimethoprim	UV UV/H2O2
	STP sludge	Narumiya et al. 2013 [47]	LC-MS/MS	NA	NA	Clarithromycin Roxithromycin Erythrimycin Norfloxiacin Ofloxacin Sulfamethoxazole Trimethoprim	NA
		Motoyama et al. 2011 [49]	LC-MS/MS	NA	NA	Erythromycin Ciprofloxacin Levofloxacin Sulfadimethoxine Sulfamethoxazole Sulfamonomethoxine Oxytetracycline Chlortetracycline Trimethoprim	NA
	STP influent/effluent	Ghosh et al. 2009 [50]	LC-MS/MS	NA	NA	Azithromycin Clarithromycin Roxithromycin Ciprofloxacin Enrofloxacin Levofloxacin Nalidixic acid Norfloxacin Sulfadimethoxine Sulfadimizine Sulfamerazine Sulfamethoxazole Sulfamonomethoxine Tetracycline Lincomycin Salinomycin Trimethoprim	CAS AO A2O
		Okuda et al. 2008 [52]	LC-MS/MS	NA	NA	NA	CAS Biollogical nutrient removal Ozonation Chlorination
		Nakada et al. 2007 [53]	LC-MS/MS	NA	NA	Azithromycin Clarithromycin Roxithromycin Dehydrated erythrimycin Sulfapyridine Sulfamethoxazole Trimethoprim	AS Sand filtration Ozonation
	STP influent/effluent/sludge	Matsuo et al. 2011 [48]	LC-MS/MS	NA	NA	Ampicillin Amoxicillin Sulfadimethoxine Sulfamethoxazole Sulfamonomethoxine Trimethoprim	AS
		Nakada et al. 2010 [55]	GC/MS	NA	NA	Triclosan	AS
		Yasojima et al. 2006 [54]	LC-MS/MS	NA	NA	Azithromycin Clarithromycin Levofloxacin	AS
	River water	Murata et al. 2011 [8]	LC-MS/MS	NA	NA	Azithromycin Clarithromycin Roxithromycin Dehydrated Erythrimycin Sulfapyridine Sulfadimethoxine Sulfamethoxazole Trimethoprim	NA
AMR Anti- microbials	STP influent/effluent Hospital effluent River water	Azuma et al. 2019 [46]	Culture-based LC-MS/MS	CRE ESBL MDRA MDRP MRSA VRE	NA	Azithromycin Cefdinir Ciprofloxacin Clarithromycin Levofloxacin	CAS or A2O +Chrolination or Ozonation

WGS*: Comprehensive antimicrobial resistance genes were identified by whole-genome sequencing. Abbreviations: AMR, antimicrobial resistance; AO, Anaerobic-oxic; AS, activated sludge; A2O, Anaerobic-anoxic-oxic; CAS, conventional activated sludge; CRE, carbapenem-resistant *Enterobacteriaceae*; ESBL, extended-spectrum β-lactamase; MDRA, multidrug-resistant *Acinetobacter baumannii*; MDRP, multidrug-resistant *Pseudomonas aeruginosa*; MRSA, methicillin-resistant *Staphylococcus aureus*; NA, not applicable; PCR, polymerase chain reaction; STP, sewage treatment plant; UV, ultraviolet; VRE, vancomycin-resistant enterococci.

### 3.2. AMR in Wastewater in Japan

#### 3.2.1. AMR in STP Influent

Suzuki et al. [43] detected carbapenem-resistant *Klebsiella quasipneumoniae* subsp. *quasipneumoniae* in urban STP influent; the bacterium harbors IncA/C2 plasmid and carries the carbapenemase gene *bla*_KHM-1_. Tanaka et al. [33] analyzed ESBL-producing *E. coli* strains detected in STP influent and showed that clinically relevant strains belonging to global epidemic clone B2-ST131, which carry global dominant ESBL genes such as *bla*_CTX-M-14_, *bla*_CTX-M-15_, and *bla*_CTX-M-27_, were widely distributed in sewage in Japan. Azuma et al. [27,46] and Eda et al. [59] detected the clinically important ARB, including ESBL-producing bacteria, carbapenem-resistant *Enterobecteriaceae* (CRE), multidrug-resistant *Pseudomonas aeruginosa* (MDRP), multi-drug-resistant *Acinetobacter* (MDRA), methicillin-resistant *Staphylococcus aureus* (MRSA), and vancomycin-resistant enterococci (VRE), in STP influent. Azuma et al. also showed that the concentrations of ARB were 206 to 653 colony-forming units per mL (CFU/mL) for CRE, 825 to 1923 CFU/mL for ESBL, 174 to 1829 CFU/mL for MDRA, 41 to 978 CFU/mL for MDRP, 35 to 892 CFU/mL for MRSA, and 34 to 916 CFU/mL for VRE and that these concentrations were 1/10 to 1/1,000,000 lower than those in LMICs [17,46,63,64,65]. The above findings show that STP influent in Japan contains a wide variety of ARB but that the concentrations are quite low compared with those in LMICs; this difference is probably due to the lower prevalence of these ARB in people in Japan than in people in LMICs [63,64,65,66].

#### 3.2.2. AMR in STP Effluent

Azuma et al. [27,46] reported the presence of ESBL-producing bacteria, CRE, MDRP, MDRA, MRSA, and VRE in STP effluent and river water, although the concentration was significantly reduced to 1/10 to 1/100 of that in the STP influent after the CAS process followed by additional disinfection treatments with chlorine or ozone. Similarly, other studies detected ARB, including ESBL and carbapenemase-producing bacteria, in STP effluent in Japan [30,32,36,41,42,44], but the ARB concentration was quite low, i.e., at the level of not detected (ND) to several CFU/mL [27,46].

It is unclear whether such a low concentration of ARB could affect the water quality of rivers, i.e., whether it would make them a source of AMR. Yamashita et al. [35] investigated the resistance to several antimicrobials, including ampicillin, first- to fourth-generation cephalosporins, imipenem, aminoglycosides, quinolones, and sulfamethoxazole/trimethoprim, in *E. coli* strains isolated from river water before and after the confluence of STP effluent; the group found that the concentration of *E. coli* and the percentage of strains that were resistant to one or more of the antimicrobials was higher in the effluent than in the river water but that the concentration and the percentage in the river water increased after the confluence of the effluent. Similar to the above studies, Iwane et al. [38] reported identifying *E. coli* strains resistant to one or more of ampicillin, cephalothin, kanamycin, gentamicin, tetracycline, chloramphenicol, and nalidixic acid in STP effluent and river water. Nishiyama et al. [62] reported the presence in both river water and STP effluent of VRE harboring the *vanC* genes. These articles suggest that various ARBs are present in STP effluent in Japan and that they represent a loading source of ARB in river water.

Furukawa et al. [56] detected the *vanA* and *vanB* in water samples collected at each stage of the CAS and chlorination processes and in STP effluent, and Liu et al. [39] showed that STP effluent increased the amounts of ARGs, including *bla*_TEM_, *tetA*, *tetW*, *sul1*, *ereA*, and *qnrD* in river water. Suzuki et al. [61] examined the periphyton (algae and algae/bacteria biofilms) in river sediment and the sediment itself and concluded that they were a hotspot for ARGs because *E. coli* strains resistant to one or more antimicrobials, including ampicillin, cefazolin, cefotaxime, and ciprofloxacin, were abundant in the periphyton and sediment and that the drug-resistant *E. coli* strains were genetically close to other non-resistant *E. coli* strains detected in the same periphyton and sediment. These reports showed that ARGs in sewage are not degraded by STP treatment and could be an important source of ARGs in river water [39,56].

#### 3.2.3. AMR in STP Sludge

AS, which is produced during the wastewater treatment process and used to treat wastewater, contains a dense microbial community and is known to be an important reservoir that maintains STP levels of ARB and ARGs, including multidrug-resistant fecal coliforms such as ESBL-producing *E. coli* and MRSA with the molecular characteristics of hospital-acquired MRSA [29,67,68].

We found only a limited number of articles on studies that examined AMR in AS in Japan. Residual ARB and ARGs in AS were reported by Miura et al. (β-lactam-antibiotic-resistant *Acidovorax* sp.) [45] and Mori et al. (bleomycin-resistance genes) [40]. Honda et al. [29] evaluated the resistance to amoxicillin, ciprofloxacin, norfloxacin, kanamycin, sulfamethoxazole/trimethoprim, and tetracycline of *E. coli* sampled at each stage of the AS process (STP influent, primary treatment effluent, AS, return sludge, and secondary treatment effluent) at an STP. Among those stages, the AS and return sludge had the highest abundance of drug-resistant *E. coli* strains, and the abundance of such strains was higher in secondary treatment effluent than in influent, although the total population of *E. coli* was lower. Suwa et al. [37] reported similar results regarding an increased abundance of drug-resistant *E. coli* strains in secondary treatment effluent after the AS process. These reports suggested that AS may serve as a reservoir of ARB in STP, but few studies examined ARB other than *E. coli* or ARGs in AS. Further research on ARB and ARGs is needed to clarify how AS affects AMR in STP in Japan.

#### 3.2.4. AMR in Hospital Effluent

In both HICs and LMICs, ARB and ARGs generally are more abundant in hospital effluent than in community effluent, although the volume of hospital effluent is quite lower than that of urban communities [9,16]. One of the main sources of pathogenic ARB and hospital-related ARGs in STPs is wastewater effluent from healthcare facilities, especially hospitals, where resistant infections occur in inpatients who stay while undergoing treatment and commonly administered antimicrobials [9].

In Japan, Azuma et al. [27,46] and Eda et al. [59] detected ESBL-producing bacteria, CRE, MDRP, MDRA, MRSA, and VRE in hospital effluent as STP influents; the concentrations of these ARB were 132 to 560 CFU/mL for ESBL, 122 to 1350 CFU/mL for CRE, 84 to 245 CFU/mL for MDRP, 224 to 1805 CFU/mL for MDRA, 29 to 472 CFU/mL for MRSA, and 283 to 482 CFU/mL for VRE and were significantly lower than those in LMICs [17,46,63,64,65]. Yamashita et al. [35], Okubo et al. [34], and Suwa et al. [37] reported that hospital effluent in Japan contains a higher abundance of drug-resistant *E. coli* than river water and general wastewater do. These reports showed that hospital effluent is also an important source of clinically relevant ARB in Japan. However, our literature search found no articles on the correlation between the ARB that cause infections in hospitals, and the ARB detected in wastewater, so the direct risk of ARB in hospital effluent to humans or environments remains obscure.

### 3.3. Antimicrobials in Wastewater

Azuma et al. [46], Yasojima et al. [54], Ghosh et al. [50], and Matsuo et al. [48] investigated the presence of several antimicrobials in STPs influent in Japan and revealed that the concentrations of amoxicillin (approximately 300 to 500 ng/L), clarithromycin (228 to 4820 ng/L), and azithromycin (ND to 1347 ng/L) were 5 to 10 times higher than the mean concentration of these antimicrobials in municipal wastewater in countries around the world and that the concentrations of ciprofloxacin (37 to 231 ng/L), tetracycline (9 to 65 ng /L), sulfamethoxazole (159 to 184 ng/L), and trimethoprim (26 to 106 ng/L) were 10 to 1000 times lower [69]. Azuma et al. [46] also showed a high concentration of levofloxacin (608 to 1970 ng/L) and a low concentration of cefdinir (ND to 80 ng/L). These results may reflect the trend of antimicrobial consumption in Japan, i.e., high use of amoxicillin, levofloxacin, clarithromycin, and azithromycin and low use of cefdinir, ciprofloxacin, tetracycline, sulfamethoxazole, and trimethoprim [70]. The patterns of residual antimicrobial classes in STP influent reflects the antimicrobial consumption patterns in a country, and these consumption patterns differ between countries and are not correlated with income level [10,69].

Azuma et al. [46] and Nakada et al. [53] examined the concentration of antimicrobials in the effluent of an STP. They showed that the concentrations of ciprofloxacin (2 to 16 ng/L), sulfamethoxazole (3.66 to 39.9 ng/L), and trimethoprim (0.22 to 16.3 ng/L) were much lower the than values of the predicted no-effect concentrations (PNECs) for resistance proposed by Bengtsson-Palme and Joakim-Larsson [71], which have been used as an indicator in several studies [69]; however, the concentrations of clarithromycin (1 to 466 ng/L) and azithromycin (45 to 270 ng/L) sometimes exceeded the PNEC values. Azuma et al. [46] also showed that the concentrations in hospital effluent of clarithromycin (16 to 1387 ng/L) and azithromycin (ND to 460 ng/L) were 5 to 10 times lower in Japan than in other countries but still exceeded the respective PNEC values. Narumiya et al. [47] reported that clarithromycin and ciprofloxacin exist at high concentrations in thickened sludge (644 to 3830 ng/mL and 47 to 2390 ng/L, respectively) and digested sludge (47 to 2390 and 102 to 449 ng/L). Those antimicrobial residues that exist at levels above the PNEC values likely contribute to the selection and dissemination of AMR in STP.

In contrast, investigations of nationwide river water in Japan by Azuma et al. [46] and Murata et al. [8] revealed low concentrations of all the examined antimicrobials, including clarithromycin (ND to 233 ng/L), azithromycin (ND to 179 ng/L), levofloxacin (6 to 14 ng/L), ciprofloxacin (2 to 10 ng/L), and sulfamethoxazole (0.01 to 33.9 ng/L); all concentrations were below the respective PNEC values. The impact of those antimicrobial residues in rivers on AMR may be limited; however, only a limited number of antimicrobial classes have been assessed. The concentrations of residual antimicrobials vary widely depending on the location and timing of sampling [8,46]. Therefore, along with further efforts to reduce antimicrobials in sewage and river water, further research and continuous monitoring are needed to evaluate the levels of clinically important antimicrobials, including cephalosporins and carbapenems, in STP effluent and river water.

### 3.4. Direct Risk of AMR in Wastewater for Humans and Environments in Japan

Knowledge of the links between AMR in wastewater and in people in Japan is still scarce. Using molecular phylogenetic analysis, Ishiguro et al. [58] and Yanagimoto et al. [57] found that *Salmonella* spp. strains resistant to multiple drugs, including ampicillin, third-generation cephems, aminoglycosides, and tetracyclines, detected in STP influent had high genetic homology to clinical isolates from human feces. Gomi et al. [36] showed that the clinically important ESBL-producing *E. coli* clone ST131 C1-M27 was commonly found in both hospital wastewater and STP effluent. Since ARB derived from humans are the best candidates for transmitting AMR back to humans, these articles showed that exposure to wastewater carries a significant risk of infection by those ARB.

The main sources of AMR and antimicrobials are STP effluent discharged into rivers and seas and treated sewage sludge applied to agricultural land [9,18]. Humans can be exposed to AMR and antimicrobials from wastewater by consuming contaminated agricultural and marine products and spending time in contaminated recreational waters [9,18]. Because most of the sludge is used as landfill or organic fertilizer in Japan [23,29], there are concerns that AMR and antimicrobials in sewage sludge may contaminate the soil and crops. Recreational waters, including coastal bathing waters that may be contaminated by wastewater are increasingly recognized as a reservoir of AMR in HICs [9,72]. For example, Sekizuka et al. detected carbapenemase gene-harboring *E. coli*, *K. pneumoniae*, and *Aeromonas* spp. strains in coastal waters of Tokyo Bay, Japan [32,41,42]. Thus, AMR in wastewater can be transmitted to humans and environments through various pathways, also in Japan. However, only a few findings indicate an association between environmental AMR and humans, and the direct risk of environmental AMR to humans is still unclear. To clarify the direct risk to humans of AMR in wastewater, further epidemiological studies of ARB and ARGs in humans and environments (including landfills and agricultural soil, coastal and marine water, and other recreational waters) are needed that use molecular typing methods such as WGS to track the ARB and ARGs to specific sources, including wastewater [9].

### 3.5. Methods for Measuring and Evaluating AMR and Antimicrobials in Wastewater

Most of the examined articles in this review reported on studies that assessed AMR in wastewater by culture-based methods, which are well-formulated and standardized procedures [9], and only a few used culture-independent methods for identification of ARGs (quantified by real-time PCR [56], ultrafiltration-based DNA detection [39], and metagenomic screening [40]). ARGs in the environment play a crucial role in both the emergence and the spread of AMR because they can be spread among different bacterial species and microbial communities in different environments through horizontal gene transfer [73]. Investigations of environmental AMR that rely solely on the culture-based method may underestimate the actual amount of AMR because that method cannot detect and often does not target non-culturable and non-pathogenic ARB, which are also active reservoirs of ARGs, and extracellular DNA [74]. Although no well-standardized procedures exist, culture-independent methods are more useful for detecting ARGs than culture-dependent methods, regardless of the viability of targets [7]. Therefore, in addition to culture-based methods, in the future, AMR investigation in water environments in Japan needs to be performed with culture-independent methods.

In most of the reviewed articles, LC-MS was used to measure antimicrobial residues in wastewater [8,46,47,48,49,50,51,52,53,54]. LC-MS is commonly used to detect antimicrobials in water because of its high selectivity and sensitivity [9]. However, when applying the LC-MS method, researchers should be aware that the results are affected by other sample components, such as dissolved organic compounds, proteins, and fatty acids, and that LC-MS has no standardized procedure, even though published methods for the analysis of antimicrobial residues provide robust validation data [9]. Standardized methods need to be established to allow comparisons of antimicrobial concentrations in sewage at different locations and times across countries.

### 3.6. Effectiveness of Risk Mitigation Interventions

Japan has one of the highest sewage coverage rates in the world [26], and most human wastewater is treated at STPs before being discharged into the environment; however, Japan is currently facing the socioeconomic problem of a rapidly shrinking population and a consequent reduction in public spending [23]. There is an urgent need to develop and apply more effective and cost-efficient technologies for wastewater treatment systems.

Wastewater treatment processes in STPs vary widely in their ability to reduce ARB and ARGs [9]. In Japan, most of the centralized STPs use a combination of CAS and disinfection processes, most of which use chlorination and some of which use UV disinfection or ozonation [23]. These processes effectively reduce ARB in wastewater, albeit not completely [16,27,46,75]. Azuma et al. [27,46]. showed that chlorination removes ESBL-producing *enterobacteriaceae*, MDRA, MRSA, and VRE less efficiently than other bacteria. *Staphylococci* and *E. coli* show resistance to ozonation [9], and ARGs in wastewater are difficult to remove by those treatments [39,56]. Furthermore, chlorination and ozonation may produce carcinogenic byproducts when water is reused, and ozonation is costly. UV disinfection has the advantage that it is compact, does not produce chemical residues, and is effective in reducing ARGs; however, it can cause regrowth and photoreactivation of pathogenic bacteria [76]. Currently, various new technologies, including a membrane-separation process, are being developed to improve the ability of STPs to reduce levels of ARB and ARGs [9,76,77]. The membrane-separation process has shown the most promising results in reducing both ARB and ARGs, although it is expensive [9].

To reduce the burden on STPs, pre-treatment of wastewater at its source before it enters the sewage system has attracted much attention [9]. Because the cost of treating wastewater is based on the volume of waste treated, source treatment is a cost-effective strategy because the treated volumes are much lower [9]. Therefore, on-site disinfection of hospital wastewater might be a promising strategy to reduce AMR because hospital wastewater is an important source of AMR not only in Japan but also in other HICs and in LMICs [9,17,27,34,35,36,37,46,59,63,64,65,78,79]. Studies are needed to examine the significance and feasibility of introducing these new technologies and strategies for developing and improving STPs in Japan.

The efficiency of STPs in removing antimicrobials from wastewater varies depending on the type of antimicrobials and the treatment method, e.g., the CAS process shows poor removal rates for trimethoprim, macrolides, and quinolones compared with penicillins and sulfamethoxazole [47,48,53,54]. Among the AS processes, the anaerobic-aerobic and anaerobic-anoxic-aerobic processes have a higher removal rate of residual antimicrobials than the CAS process [50]. The combination of CAS with sand filtration and ozonation significantly increases the removal rate compared with CAS alone [53]. Kim et al. [51] showed that macrolide antimicrobials, including erythromycin, clarithromycin, and azithromycin, were not degraded effectively during the UV process, even though the process used a considerably higher UV dose than is usually used for water disinfection. Although there are differences in the efficacy of wastewater treatment methods in removing antimicrobials, the complete removal of antimicrobials from wastewater is difficult by any of the above methods.

Our review suggests that most of the residual antimicrobials in sewage water in Japan are antimicrobials administered to hospital inpatients rather than those consumed by outpatients at home. Azuma et al. [46] compared the concentration of antimicrobial residues in hospital effluent with that in STP effluent and found that the concentration in the former was low and that most of the antimicrobials in STP effluent were from outside the hospital. In Japan, oral antimicrobials prescribed for an outpatient account for 92.4% of the total antimicrobials used daily, and most of them are used inappropriately [80]. Therefore, to reduce the burden of antimicrobials on the environment, efforts should be made to ensure that patients use oral antimicrobials correctly and that new technologies and strategies to treat antimicrobial agents in STP are developed and introduced.

## 4. Conclusions

Our review revealed that clinically important ARB, including ESBL-producing bacteria, CRE, MDRP, MDRA, MRSA, and VRE, and antimicrobial residues are present in human wastewater in Japan. The concentrations of the ARB and antimicrobials are quite low in Japan compared with those in LMICs because of the low prevalence of AMR, and they are further reduced by STP treatment before the wastewater is discharged into the river water. Nevertheless, the remaining ARB, along with ARGs (which cannot be degraded by STP treatment), could be an important source of contamination in river water. AS in STP contains various ARB, including drug-resistant *E. coli* and ARGs, and can act as a reservoir that maintains levels of AMR in STP. Hospital effluent also contains clinically important ARB, as listed above, and can be an important source of them. The persistence in wastewater of antimicrobial residues above the PNEC values in wastewater may contribute to the selection and dissemination of AMR in the Japanese sewage treatment system. AMR in wastewater may affect humans and environments through direct exposure or indirect pathways through contaminated sludge. Because most of the studies used culture-based methods and almost none used culture-independent methods to assess the AMR in wastewater, ARGs in wastewater were poorly evaluated in Japan. The CAS process in combination with chlorination and ozonation, as commonly used in Japanese STPs, may be inadequate for completely removing AMR and antimicrobials in wastewater; therefore, to reduce the burden of AMR and antimicrobials on wastewater treatment systems, new strategies to treat AMR and antimicrobial agents in STP, including on-site source treatment and UV processes, need to be developed and introduced, and the proper use of oral antimicrobials needs to be ensured.

This study has several limitations. First, because each article included in this review analyzed different types of ARB/ARGs and antimicrobials in different wastewater sources at different times with different methods, it is difficult to compare the data between articles. Second, no articles assessed the relationship between the presence of antimicrobials and the occurrence of AMR in wastewater. Third, none of the articles described the actual impact of AMR and antimicrobials in wastewater on the environment or the direct risk to human health in Japan, so which interventions effectively reduce AMR and antimicrobials in wastewater remains unclear. Nevertheless, our review showed the importance of the monitoring for AMR and antimicrobials in STP influent/effluent, STP sludge, river water, and hospital effluent and of the efforts to reduce the contamination load of AMR and antimicrobials in wastewater before it is discharged into the environment. Future actions and research to develop strategies to reduce the levels of AMR and antimicrobials in wastewater in HICs are warranted.

## Figures and Tables

**Figure 1 antibiotics-11-00849-f001:**
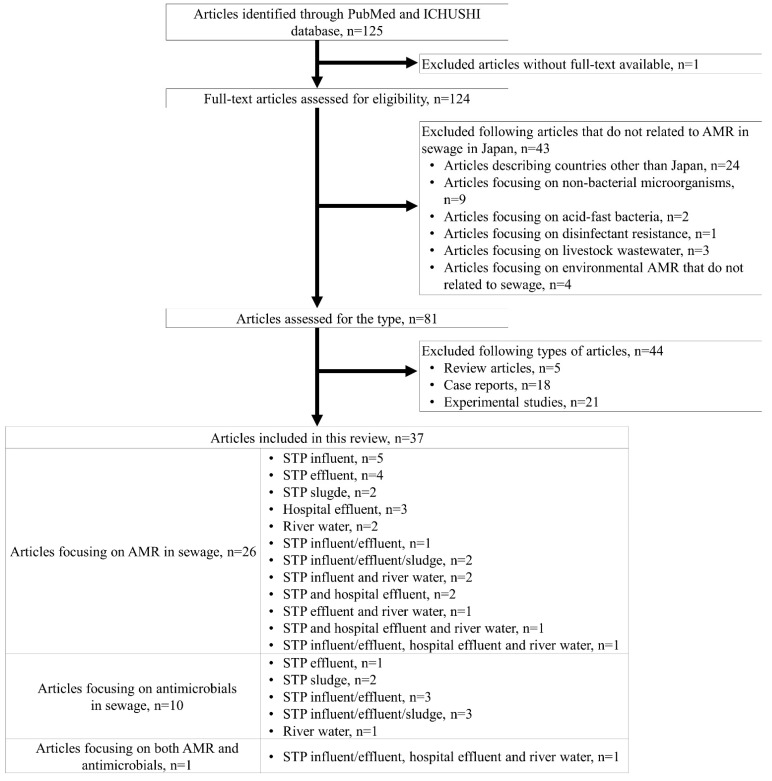
Flow chart of the review process and article screening results. Abbreviations: ICHUSHI, Igaku Chuo Zashi; AMR, antimicrobial resistance.

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
