# Peer review of "Review of Antimicrobial Resistance in Wastewater in Japan: Current Challenges and Future Perspectives"

_antibiotics, 2022, doi:10.3390/antibiotics11070849_

Round 1
Reviewer 1 Report
The breakdown of the article categories reviewed, based on what is excluded or included, is organized.
However, a discussion on the complete list of exclusion/inclusion criteria needs to be presented prior to listing the results of the literature search. The reader needs to know what you are looking for and why.
It is best, but not required, to follow a process of evaluating literature articles such as in a "systematic review" that is composed of a series of evaluations in the form of developing a protocol, to critical appraisals of the articles that initially fulfill the right categories, and eventually a complete systematic review.
Author Response
Our responses to the comments from Reviewer 1:
C1-1 The breakdown of the article categories reviewed, based on what is excluded or included, is organized.
However, a discussion on the complete list of exclusion/inclusion criteria needs to be presented prior to listing the results of the literature search. The reader needs to know what you are looking for and why.
It is best, but not required, to follow a process of evaluating literature articles such as in a "systematic review" that is composed of a series of evaluations in the form of developing a protocol, to critical appraisals of the articles that initially fulfill the right categories, and eventually a complete systematic review.
R1-1 We thank the reviewer for this suggestion. For this review, we focused on antimicrobial resistance (AMR) in bacteria other than acid-fast bacteria. We excluded articles on non-bacterial microorganisms including fungi and viruses, and did not consider disinfectant resistance. Although agricultural wastewater and industrial discharge are considered also to be important reservoirs of AMR in other countries, our review focused only on sewage because research has shown that AMR in river water in Japan originates mainly from human waste. We added this discussion on the exclusion/inclusion criteria on p.5, lines 93-97.
Reviewer 2 Report
This review is sound and easy to read. Recent references are included. The main part of the paper is a large table with information on each paper evaluated. Here it becomes clear that nearly each paper has a different aim. Different types of bacteria researched, different methods used, different types of wastewater analyzed, which makes comparisons really difficult. This observation is also valuable for the conclusions which remain superficial.
I think time is not really ready for a conclusive review on this important topic.
Author Response
Our responses to the comments from Reviewer 2:
C2-1 This review is sound and easy to read. Recent references are included. The main part of the paper is a large table with information on each paper evaluated. Here it becomes clear that nearly each paper has a different aim. Different types of bacteria researched, different methods used, different types of wastewater analyzed, which makes comparisons really difficult.
R2-1 We appreciate the reviewer’s comment on this point. Because each article included in this review analyzed different types of antibiotic resistant bacteria (ARB)/antibiotic resistant genes (ARGs) and antimicrobials in different wastewater sources at different times with different methods, it is difficult to compare the data between articles. Nevertheless, our review showed the importance of the monitoring for AMR and antimicrobials in sewage treatment plant (STP) influent/effluent, STP sludge, river water, and hospital effluent and of the efforts to reduce the contamination load of AMR and antimicrobials in wastewater before it is discharged into the environment. We added this on p.21, lines 409-418.
C2-2 This observation is also valuable for the conclusions which remain superficial.
I think time is not really ready for a conclusive review on this important topic.
R2-2 Thank you for providing these insights. We provided a conclusion for each of the discussion. Please see p.7, lines 133-135, p.8, lines 148-151, p.9, lines 172-174, p.15, lines 305-315, p.17, lines 337-338, and p.19, lines 376-378.
Reviewer 3 Report
This article aimed to review the occurrence of antimicrobial resistance (AMR) in wastewater in Japan. The major concern is that major texts and the table in this article are more like piles of reference results and did not provide in-depth discussion or conclusions. A few typos and grammar mistakes were also being presented. Detailed comments and suggestions are listed below.
1. Introduction.
1) “The residual antimicrobials in wastewater that are used in humans, animals, and plants”, what does “plants” refer to?
2) The ARB should be defined properly, which is short for antibiotic resistant bacteria.
3) “To date, the levels of ARB and ARGs that poses a risk to human health unknown”, a verb is missing.
3. Results and discussion
4) Table. How about the abundance of AMR?
5) Have the authors made comparison with the occurrence of AMR in wastewater from other countries?
6) Any relationship between the antimicrobials and AMR in wastewater?
7) “3.5. Methods fo measuring and evaluating AMR”, typo “fo”.
Author Response
Our responses to the comments from Reviewer 3:
C3-1 This article aimed to review the occurrence of antimicrobial resistance (AMR) in wastewater in Japan. The major concern is that major texts and the table in this article are more like piles of reference results and did not provide in-depth discussion or conclusions.
R3-1 Thank you for providing these insights. We provided a conclusion for each of the discussion. Please see p.7, lines 133-135, p.8, lines 148-151, p.9, lines 172-174, p.15, lines 305-315, p.17, lines 337-338, and p.19, lines 376-378.
A few typos and grammar mistakes were also being presented. Detailed comments and suggestions are listed below.
- Introduction.
C3-2 “The residual antimicrobials in wastewater that are used in humans, animals, and plants”, what does “plants” refer to?
R3-2 Plants refers to crop plants for which antimicrobials are widely used [Aga D, et al. Wellcome Trust. London, England, 2018. Available online: https://wellcome.org/sites/default/files/antimicrobial-resistance-environment-report.pdf]. We added this on p.3, lines 53-56.
C3-3 The ARB should be defined properly, which is short for antibiotic resistant bacteria.
R3-3 We corrected this error throughout our article.
C3-4 “To date, the levels of ARB and ARGs that poses a risk to human health unknown”, a verb is missing.
R3-4 We corrected this error.
- Results and discussion
C3-5 Table. How about the abundance of AMR?
R3-5 There were few articles that measured the abundance of AMR. Azuma et al. showed that the concentrations of ARB in STP influent were 206 to 653 colony-forming units per mL (CFU/mL) for carbapenem-resistant Enterobecteriaceae (CRE), 825 to 1,923 CFU/mL for extended-spectrum β-lactamase (ESBL), 174 to 1,829 CFU/mL for multi-drug-resistant Acinetobacter (MDRA), 41 to 978 CFU/mL for multidrug-resistant Pseudomonas aeruginosa (MDRP), 35 to 892 CFU/mL for methicillin-resistant Staphylococcus aureus (MRSA), and 34 to 916 CFU/mL for vancomycin-resistant enterococci (VRE) [Azuma et al. Sci Total Environ 2019]. The ARB concentration in STP effluent was at the level of not detected (ND) to several CFU/mL. The concentrations of ARB in hospital effluent were 132 to 560 CFU/mL for ESBL, 122 to 1,350 CFU/mL for CRE, 84 to 245 CFU/mL for MDRP, 224 to 1,805 CFU/mL for MDRA, 29 to 472 CFU/mL for MRSA, and 283 to 482 CFU/mL for VRE. We added those on p.8, lines 164-171, p.9, lines 180-182, and p.12, lines 233-237.
C3-6 Have the authors made comparison with the occurrence of AMR in wastewater from other countries?
R3-6 Among the articles we reviewed, Azuma et al. measured ARB in sewage in Japan and compared it with that of other countries [Azuma et al. Sci Total Environ 2019]. The concentrations of ARB in both STP influent and hospital effluent in Japan were 1/10 to 1/1,000,000 lower than those in Low- and middle-income countries. We added this on p.8, lines 164-171 and p.11, lines 233-237.
C3-7 Any relationship between the antimicrobials and AMR in wastewater?
R3-7 No articles assessed the relationship between the presence of antimicrobials and the occurrence of AMR in wastewater. We added this on p.21, lines 411-412.
C3-7 “3.5. Methods fo measuring and evaluating AMR”, typo “fo”.
R3-7 We corrected this error.
Round 2
Reviewer 2 Report
With some minor changes your manuscript came now back to me. Your manuscript reflects the current state of knowledge in the field.
But the bacteria researched in the different types of samples are often different, the methods used are also different and the amount of results is not very abundant. So we think that it is necessary to wait for some further results and in one year or two you can have an excellent paper.
The results presented today do not allow a real discussion and mainly a valuable conclusion.
Author Response
Our responses to the comments from Reviewer 1:
C1-1 With some minor changes your manuscript came now back to me. Your manuscript reflects the current state of knowledge in the field.
But the bacteria researched in the different types of samples are often different, the methods used are also different and the amount of results is not very abundant. So we think that it is necessary to wait for some further results and in one year or two you can have an excellent paper.
The results presented today do not allow a real discussion and mainly a valuable conclusion.
R1-1 We appreciate the reviewer’s comment on this point. The purpose of this literature review is to understand the current knowledge about and challenges related to AMR in wastewater in Japan and to provide a perspective on future research on that.
Our review showed that most of the studies that assessed antimicrobial resistance (AMR) in wastewater were examined by culture-based methods, which are well-formulated and standardized procedures, and only a few used culture-independent methods for identification of ARGs. Investigations of environmental AMR that rely solely on the culture-based method may underestimate the actual amount of AMR because that method cannot detect and often does not target non-culturable and non-pathogenic ARB, which are also active reservoirs of ARGs, and extracellular DNA. Although no well-standardized procedures exist, culture-independent methods are more useful for detecting ARGs than culture-dependent methods, regardless of the viability of targets. Therefore, in addition to culture-based methods, in the future AMR investigation in water environments in Japan need to be performed with culture-independent methods (Please see p.16, lines 292-304). Our review also showed that further epidemiological studies of ARB and ARGs in humans and environments (including landfills and agricultural soil, coastal and marine water, and other recreational waters) are needed that use molecular typing methods such as whole genome sequence to track the ARB and ARGs to specific sources, including wastewater (Please see p.15, lines 274-289).
Since our findings are based on a literature review of the past 30 years, we think it is unlikely that there will be dramatic progress in understanding of AMR in wastewater in Japan over next one year or two.
Reviewer 3 Report
The full name of "ARB" has been corrected. But the "ARGs" should be short for "antibiotic resistance genes" rather than "antibiotic resistant genes".
Author Response
Our responses to the comments from Reviewer 3:
C3-1 The full name of "ARB" has been corrected. But the "ARGs" should be short for "antibiotic resistance genes" rather than "antibiotic resistant genes".
R3-1 We thank the reviewer for pointing out this error. We corrected this throughout our article.